# Immune System Deficiencies Do Not Alter SARS-CoV-2 Evolutionary Rate but Favour the Emergence of Mutations by Extending Viral Persistence

**DOI:** 10.3390/v16030447

**Published:** 2024-03-13

**Authors:** Laura Manuto, Martina Bado, Marco Cola, Elena Vanzo, Maria Antonello, Giorgia Mazzotti, Monia Pacenti, Giampaolo Cordioli, Lolita Sasset, Anna Maria Cattelan, Stefano Toppo, Enrico Lavezzo

**Affiliations:** 1Department of Molecular Medicine, DMM, University of Padova, 35121 Padova, Italy; laura.manuto@phd.unipd.it (L.M.); martina.bado@studenti.unipd.it (M.B.); elena.vanzo.1@studenti.unipd.it (E.V.); maria.antonello@studenti.unipd.it (M.A.); giorgia.mazzotti@studenti.unipd.it (G.M.); giampaolo.cordioli@studenti.unipd.it (G.C.); annamaria.cattelan@aopd.veneto.it (A.M.C.); 2Department of Medicine, DIMED, University of Padova, 35128 Padova, Italy; marco.cola@studenti.unipd.it; 3Unit of Microbiology and Virology, University Hospital of Padova, 35128 Padova, Italy; monia.pacenti@aopd.veneto.it; 4Unit of Infectious Diseases, University Hospital of Padova, 35128 Padova, Italy; lolita.sasset@aopd.veneto.it

**Keywords:** SARS-CoV-2 genomic variability, viral quasispecies, immunocompromised subjects, intra-host

## Abstract

During the COVID-19 pandemic, immunosuppressed patients showed prolonged SARS-CoV-2 infections, with several studies reporting the accumulation of mutations in the viral genome. The weakened immune system present in these individuals, along with the effect of antiviral therapies, are thought to create a favourable environment for intra-host viral evolution and have been linked to the emergence of new viral variants which strongly challenged containment measures and some therapeutic treatments. To assess whether impaired immunity could lead to the increased instability of viral genomes, longitudinal nasopharyngeal swabs were collected from eight immunocompromised patients and fourteen non-immunocompromised subjects, all undergoing SARS-CoV-2 infection. Intra-host viral evolution was compared between the two groups through deep sequencing, exploiting a probe-based enrichment method to minimise the possibility of artefactual mutations commonly generated in amplicon-based methods, which heavily rely on PCR amplification. Although, as expected, immunocompromised patients experienced significantly longer infections, the acquisition of novel intra-host viral mutations was similar between the two groups. Moreover, a thorough analysis of viral quasispecies showed that the variability of viral populations in the two groups is comparable not only at the consensus level, but also when considering low-frequency mutations. This study suggests that a compromised immune system alone does not affect SARS-CoV-2 within-host genomic variability.

## 1. Introduction

Since the beginning of the SARS-CoV-2 pandemic, several studies have emphasised a prolonged positivity of RT-PCR tests in the immunosuppressed population, which can be explained by the impairment of viral clearance by weakened innate and adaptive immune responses in immunocompromised patients [1]. Although a positive RT-PCR test only reflects the detection of viral RNA and does not necessarily indicate the presence of active viral replication or infectivity [2], the specific host environment of immunocompromised subjects has been suggested to be responsible for accelerated viral evolution [1,3] or to induce the selective accumulation of viral mutations in genomic regions that affect viral fitness, promoting immune system escape and transmission [4,5,6,7]. In addition to the compromised immune system, typical treatments, such as convalescent plasma, monoclonal antibodies or antivirals, have also been reported to induce selective pressure on SARS-CoV-2 intra-host evolution [5,8,9,10,11]. Moreover, protracted SARS-CoV-2 infections in immunocompromised subjects have been indicated as a possible source for the origin of novel SARS-CoV-2 variants, namely B.1.1.7 (Alpha) [7] and of B.1.1.529 (Omicron) [6], due to the high difference in the number and location of mutations compared with the viral haplotypes circulating at the time, according to sequence availability in public databases. However, most of the available data are based on case studies of immunocompromised subjects who were monitored due to recurrent or prolonged hospitalisation, whereas very few longitudinal studies have been performed on immunocompetent subjects [12,13,14,15,16]. Nonetheless, those studies reported the longitudinal accumulation of novel SARS-CoV-2 mutations affecting viral fitness and immune escape in non-immunocompromised subjects as well. Very few studies currently available have assessed intra-host SARS-CoV-2 evolution in immunocompromised and non-immunocompromised subjects [15,16].

In this study, we used a whole-genome deep sequencing approach on serial nasopharyngeal swab samples collected from immunosuppressed and non-immunocompromised patients, treated with antiviral agents or monoclonal antibodies, with the aim of investigating SARS-CoV-2 intra-host evolution and assessing the potential role of a compromised immune system in driving the accumulation of mutations, both quantitatively and qualitatively. Samples collected the day of the first positive SARS-CoV-2 test (T0), and after 3 (T3), 7 (T7), 14 (T14), 21 (T21) and 30 (T30) days, according to infection length, were deep-sequenced and analysed for longitudinal viral haplotype variation and for low-frequency mutation diversity and shifts. Given that there are no standard guidelines for studying low-frequency variants, we decided to rely on a capture-based protocol that was recently proven to be sensitive and efficient in providing an unbiased full coverage and a reliable representation of low-frequency variants [17,18,19]. Moreover, we minimised the number of PCR cycles normally required by the protocol to avoid the generation of artefacts commonly associated with amplicon sequencing. In addition, a subset of samples was sequenced twice to allow the precise calibration of the parameters for reliably calling the low-frequency variants.

## 2. Materials and Methods

### 2.1. Patients and Samples

We conducted a single-centre prospective study at Padova University Hospital from December 2021 to April 2022. A group of eight immunocompromised and a group of fourteen non-immunocompromised adult subjects, positive for SARS-CoV-2, were enrolled among those evaluated at the outpatient clinic for COVID-19 early treatment. All participants underwent regular nasopharyngeal swabs for the detection of SARS-CoV-2 at 3, 7, 14, 21 and 30 days after the first positive test, or until negativization. The enrolled patients were prescribed antiviral therapy (remdesivir or monoclonal antibodies) in accordance with the criteria outlined in the Italian guidelines. Individuals were divided into two groups: severely immunocompromised patients (Group 1) and non-immunocompromised patients but at high risk of COVID-19 clinical progression (Group 2) [20].

The SARS-CoV-2 genome was searched with Simplexa™ COVID-19 Direct assay (DiaSorin, Sydney, Australia) targeting the S gene and ORF1ab gene, following manufacturer instructions. After inactivating nasopharyngeal swab samples at 90 °C for 30 min, total nucleic acids were purified using a MagNA Pure 96 System (Roche Applied Sciences, Penzberg, Germany) following manufacturer instructions.

The study was performed according to the ethical guidelines of the Declaration of Helsinki (7th revision). All the patients gave their written informed consent and all analyses were carried out on anonymised data as required by the Italian Data Protection Code (Legislative Decree 196/2003) and the general authorization issued by the Data Protection Authority.

### 2.2. Library Construction, and Sequencing Methods

The sequencing libraries were prepared in accordance with the protocol “Creating cDNA libraries using Twist library preparation kit for ssRNA virus detection for use with target enrichment workflow” and enriched with the Twist SARS-CoV-2 Research Panel. Samples were treated with DNAse in order to eliminate DNA and quantified with a Qubit RNA High-Sensitivity assay. RNA samples were then diluted and cDNA was synthetised and quantified with a Qubit DNA High-Sensitivity assay. Samples were enzymatically fragmented, end-repaired and dA-tailed in order to generate dA-tailed DNA fragments with an average size of 300 bp. Ligation of TWIST Universal Adapters and purification was performed. cDNA libraries were amplified with Twist Unique Dual Index Primers (10 PCR cycles), purified and quantified with a Qubit DNA High-Sensitivity assay. The resulting libraries were validated using the Fragment Analyzer (High-Sensitivity Small Fragment Analysis Kit) to check size distribution. An equal amount of the indexed libraries was pooled to reach a total mass of 1500 ng. Each pool was hybridised with SARS-CoV-2 probes and the capture was amplified (9 PCR cycles) and purified with Ampure beads. The resulting libraries were validated using the Fragment Analyzer (High-Sensitivity Small Fragment Analysis Kit) to check size distribution. The concentration of the libraries was defined on the basis of the Qubit 4.0 Fluorometer. The libraries were normalised to 4 nM and loaded at a concentration of 1,2 pM onto an Illumina Mid Output Flowcell with 1% of Phix control. The samples were then sequenced with NextSeq using an Illumina V2 chemistry 2 × 150 bp paired-end run.

### 2.3. Viral Genome Assembly: Quality Check and Mapping of the Reads

The raw sequences were filtered for length and quality with Trimmomatic [21] v0.39 according to the following parameters: ILLUMINACLIP:TruSeq3-PE-2:2:30:10, LEADING:30, TRAILING:30, SLIDINGWINDOW:4:20, and MINLEN:40. High-quality reads were aligned on the SARS-CoV-2 reference genome (genbank ACC: NC_045512) with BWA-MEM v0.7.17. Duplicated reads were then removed with Picard v3.0.0 (http://broadinstitute.github.io/picard/, accessed on 10 May 2023). Consensus sequences were generated using a combination of SAMtools [22] v1.11 and VarScan [23] v2.4.4 variant caller (http://varscan.sourceforge.net, accessed on 10 May 2023). First, all bam files were cleaned from secondary alignments, incorrectly mapped reads, low-quality reads and unpaired reads. The consensus sequences were then reconstructed by utilising VarScan with no filters to call all the identified bases and variants per each genomic position. Subsequently, the output was filtered by an in-house Python script that required a minimum of 5 reads to call a base, at least 2 forward and 2 reverse reads and a frequency greater than 50% to call a major variant, and that automatically introduced ‘Ns’ in low-quality or uncertain/uncovered regions of the reference sequence. Eventually, BCFtools [22] v1.11 was utilised to generate the consensus sequences, whereas SARS-CoV-2 clades and haplotypes were called with the NextClade [24] Web tool (https://clades.nextstrain.org, accessed on 19 October 2023). The 52 SARS-CoV-2 sequences produced in this study were submitted to the GISAID portal (www.gisaid.org, accessed on 30 November 2023) [25]. Appendix A reports the correspondence of the GISAID IDs of the newly produced sequences with the identifiers reported in this paper.

### 2.4. Low-Frequency Variants Validation and Identification

Twenty-three of the collected samples were sequenced twice to be utilised as controls for minor variants’ reproducibility. After performing the viral genome assembly as described above, we analysed all mutations recorded in the Variant Call Format (VCF) files generated using LoFreq [26] v2-1-5 without imposing any restrictions on mutation frequency or strand bias. First, a custom strand filter was applied by requiring a minimum of two forward and two reverse reads supporting each mutation, whereas variants occurring in the first and last one hundred bases were excluded from consideration. To compare the type of the minor variants identified in the replicated samples, the overlap coefficient was calculated for each couple, imposing different frequencies to call minor variants, ranging from 0.5% to 5%, considering the median coverage of each replicate (Venn diagrams provided for 1% frequency only, Appendix A).

### 2.5. Principal Component Analysis, PCA

For each group of replicates, the replicate with the highest coverage was chosen to represent the sample. In cases where a sample was sequenced only once, we considered the single available replicate. The final dataset comprised thirty samples, which are summarised in Appendix A. 

The presence of minor mutations reverting to the NC_045512 reference and occurring outside indels was manually evaluated and duly taken into account.

To assess the evolution of minor variants in the immunocompromised and healthy individuals, a principal component analysis (PCA) was conducted using the scikit-learn package (v1.3.0) in Python3. 

The dataset used for this analysis includes the list of low-frequency mutations identified in the samples along with their frequency. Before performing the PCA, the mutation frequencies were normalised using the StandardScaler module from the scikit-learn package. The contribution of variance in the resulting principal components (PCs) was also assessed and reported specifically for PC1 and PC2.

## 3. Results

### 3.1. Cohort Characterisation

A total of 22 patients were enrolled between December 2021 and April 2022, all exhibiting mild COVID-19 symptoms. Eight were classified as immunocompromised individuals (Group 1). This group consisted of three patients with active solid tumours, four patients undergoing chemotherapy for haematological malignancies, and one recent (<1 year) liver transplant recipient on immunosuppressive therapy. The remaining fourteen patients were non-immunocompromised subjects at high risk of COVID-19 clinical progression (Group 2), who presented with various underlying health conditions: three patients had chronic lung disease; six patients had heart disease; one patient had obesity (BMI > 30 kg/m^2^); one patient had type 2 diabetes mellitus; and three patients had chronic liver disease.

The immunocompromised group presented a mean age of 60 (IQR: 50.5–69), with 75% being males, whereas Group 2 subjects were characterised by a mean age of 72 (IQR: 64.3–83.5) and 50% being males.

All patients had received two vaccinations for SARS-CoV-2, except for two patients in Group 1. All Group 2 subjects received three days of remdesivir therapy, whereas, in Group 1, two patients received remdesivir, five received monoclonal antibodies, and one patient was treated with a combination of remdesivir and monoclonal therapy.

Characteristics and clinical information of the studied cohorts are summarised in Table 1. 

The length of SARS-CoV-2 infection, defined as the time window spanning from the day of the first positive molecular test to the day of the first negative test, was prolonged in immunocompromised patients (21 IQR 21–30) compared with the non-immunocompromised subjects (14 IQR: 14–14; *p* < 0.001, Mann–Whitney test).

All the swabs that yielded a positive result in the molecular test for detecting SARS-CoV-2 underwent deep sequencing. This involved employing a shotgun methodology on virus-enriched nucleic acids, obtained with SARS-CoV-2-specific probes. In total, 26 and 26 viral full genomes were obtained from the immunocompromised and non-immunocompromised patients, respectively.

Infection lengths of all the subjects and the timepoints for which full-coverage viral sequences were available are summarised in Figure 1. The emergence of mutations over time was monitored at both the consensus level (a mutation must be present in more than 50% of reads covering that position) and the quasispecies level (mutations equal or below the 50% frequency threshold).

### 3.2. Intra-Host Variation of SARS-CoV-2 Consensus Genome in Immunocompromised and Non-immunocompromised Subjects

SARS-CoV-2 intra-host evolution was investigated in seven immunocompromised and seven non-immunocompromised subjects, according to the availability of viral full-genome sequences from multiple timepoints, including at least T0 (the day of the first positive swab) and T7 (the seventh day after the first positive swab). Details about SARS-CoV-2 lineage, number and type of nucleotide and amino acid mutations, together with the timepoint at which the novel mutations were observed, are summarised in Table 2.

As shown in Figure 2, mutations in the viral genome occurred in three (42.85%) immunocompromised subjects and in two non-immunocompromised (28.57%) subjects. No significant difference in the number of subjects who experienced intra-host viral evolution was observed between the two groups (Fisher exact test, *p* = 1). By considering the same time window, including the timepoints T0 and T7, two subjects out of seven acquired novel viral mutations in both groups, further confirming the lack of difference according to the immune system state. 

The immunocompromised subject I_4, a 71-year-old man with pulmonary squamous cell carcinoma, was treated with remdesivir. He was infected with SARS-CoV-2 lineage AY.43 (Delta) [27] and the infection lasted 30 days. At 3 days after infection, a non-synonymous mutation, 4822A > C (ORF1a:Q1519H), became prevalent and persisted until the end of the infection. Notably, the mutation was observed also at T0, but it was slightly below the consensus threshold of 50%, and its frequency increased over time, eventually reaching 100% at T7 (T0: 49.96%, T3: 50.62, T7: 100%). It is unclear whether this mutation was already present as a quasispecies at the time of infection, or emerged during the early stages of viral replication in this patient. At 7 days after COVID-19 diagnosis, the virus acquired a single-nucleotide mutation in the spike gene (22821A > C), resulting in an amino acid change from Aspartic acid to Alanine in position 420 (S: D420A), which was predicted to induce an escaping advantage from the monoclonal antibody LY-CoV016 [28]. At T0, T3 and T7, the frequency of 22821A > C mutation was 0%, 0% and 100%, respectively.

The immunocompromised subject I_5, a 49-year-old man under immunosuppressive therapy due to liver transplant, was treated with bamlanivimab and etesivimab. He was infected with SARS-CoV-2 lineage BA.1 (Omicron) in February 2021 and the infection persisted for 30 days. Full-coverage sequences were available after 3, 7, 14 and 21 days, allowing detailed monitoring of intra-host viral evolution. As a result, for the first two weeks from diagnosis, no novel mutations were fixed in the prevailing haplotype, whereas on day 21 a six-nucleotide deletion became prevalent, affecting genomic positions 519–524 and resulting in the deletion of two amino acid residues and one amino acidic change (ORF1a:V84_M85del, ORF1a: E87K). The mutation was not detected at T0, T3 and T7 (0%); it appeared at timepoint 14 at low frequency (6.67%) and was fixed in the consensus sequence at timepoint 21 (53.85% frequency). Interestingly, 519_524del was only one of the several similar deletions that increased in frequency with time in this subject. In fact, as shown in Figure 3 and summarised in Table 3, six different deletions occurring within the viral genomic region 508–524 were competing from timepoint 14 onwards, with only one of them eventually prevailing in the subsequent timepoints analysed. In particular, a deletion of 15 nucleotides (508_522del) was detected in 15.44% of reads mapped in that region, and a 9-nucleotide deletion, 510_518del, was present with a frequency of 8.20%. Moreover, a six-nucleotide deletion (515_520del) was present with a frequency of 6.49%, and two three-nucleotide deletions, 516_518del and 518_520del, were present with frequencies of 1.24% and of 1.57%, respectively. These deletions occurred within the Nsp1 coding region and have been reported in the literature to correlate with lower viral load and lower serum IFN-β [29].

The third immunocompromised subject to acquire novel viral mutations over time was the subject I_7, a 55-year-old man previously diagnosed with follicular lymphoma. He was infected with SARS-CoV-2 lineage BA.1.1 (Omicron) and treated with the monoclonal antibody sotrovimab. After 7 days from the first positive test, subject I_7 acquired two novel mutations, one being silent (4012C > A) and the second one inducing an amino acid change in ORF1a: 9810C > A, resulting in ORF1a:T3182N. Interestingly, the 4012C > A mutation was not present at T0, but appeared after 3 days from infection, with a frequency of 42.03%, and reached a frequency of 100% at day 7. Similarly, 9810C > A appeared at T3 with a frequency of 1.43%, and was fixed at T7 with a frequency of 87.50%. SARS-CoV-2 was cleared after 21 days.

The non-immunocompromised subjects displaying intra-host viral evolution were subject H_6 and subject H_9; both subjects were affected by heart disease and cleared the infection within 14 days.

Subject H_6, an 83-year-old man, was infected with the BA.2 SARS-CoV-2 lineage (Omicron) and was treated with remdesivir. At day 7, he acquired the synonymous mutation 14602T > C and a deletion of nine nucleotides in ORF1a, 510_518del, resulting in three amino acid deletions and one amino acid change: ORF1a: G82_V84del and ORF1a: M85V, respectively. None of the described mutations were present at day 0 at minor frequencies, while at day 7 they were present with frequencies of 75.76% and of 83.33%, respectively.

The subject H_9, an 89-year-old man, was infected with the BA.2.9 SARS-CoV-2 lineage (Omicron) and was treated with remdesivir. At day 7, we observed the reversion of a major synonymous mutation, 25603C > T, that was present at 61.69% at day 0, with the remaining reads supporting the SARS-CoV-2 reference consensus, and that dropped to 17.62% by day 7, with the SARS-CoV-2 reference base prevailing over it (82.77%).

### 3.3. Frequency of Emerged Non-Synonymous Mutations in GISAID Database

To investigate whether the non-synonymous mutations that emerged in this study cohort had already been reported elsewhere, either before or after, all the human SARS-CoV-2 complete and high-coverage sequences available in GISAID from the beginning of the pandemic to the 15th of October 2023 were downloaded and analysed. Although at very low frequencies, all the investigated mutations were already reported in the GISAID database in periods preceding the sample collection for this study (Appendix A). However, the 510_518del was the most frequent and its frequency showed periodic peaks, with the highest peak being reported in September 2023 (Appendix A). 

### 3.4. Analysis of Viral Quasispecies

After evaluating the differences between consensus mutations emerging in the immunocompromised and in the non-immunocompromised subjects, we explored the variation of mutations appearing in the two cohorts at frequencies below or equal to 50%. Prior to performing any comparison between the groups, we defined the parameters for minor variants to be reliable and reproducible, thanks to the availability of deeply sequenced technical replicates of several samples. 

#### 3.4.1. Validation of the Detection Pipeline for Low-Frequency Mutations

Twenty-three of the collected samples were sequenced twice to be utilised as controls for minor variants’ reproducibility. All mutations recorded in the Variant Call Format (VCF) files, generated using LoFreq v2-1-5 without imposing any restrictions on mutation frequency or strand bias, were analysed. To compare the minor variants identified in the technical replicates, the overlap coefficient was calculated, imposing different minimum frequencies for minor variant calling, ranging from 0.5% to 5% (Appendix A, provided applying 1% frequency to call minor variants). In addition, we assessed the impact of the sequencing depth on the overlap coefficient (Appendix A). Finally, we investigated the potential correlation between the number of minor variants observed in each sample and the median sequencing depth (Appendix A). Although a strong correlation was observed at low depths (r_Spearman_ = 0.8, *p* = 0.0009), for samples with median sequencing depth equal to or higher than 600 reads per genomic position, the two variables were independent (r_Spearman_ = −0.30, *p* = 0.16). As a result, we defined the best combination of the minimum frequency and median sequencing depth parameters for reproducible minor variant calling, being 1% and 600 reads per genomic position, respectively. Accordingly, only samples satisfying this requirement were considered for minor variant analysis, and only mutations with a frequency between 1% and 50% included were contemplated.

#### 3.4.2. Quantitative Analysis of Minor Variants

Given that some subjects were infected with Omicron, whereas others were infected with Delta, we checked the potential impact of different viral variants in the number of detected low-frequency mutations. The analysis confirmed that no difference can be attributed to viral variants, as shown in the comparison between the Omicron and Delta viral samples of immunocompromised subjects at T0 (Appendix A). Moreover, we investigated the impact of age (Appendix A) and treatments (Appendix A) on the amounts of minor variants. Accordingly, we proceeded by accounting the immunological status as the only variable to be assessed, regardless of the SARS-CoV-2 variant of infection. To investigate whether the immunocompromised subjects were more prone to developing novel mutations, we compared the number of minor variants observed overall at T0 and at T7 in the immunocompromised and in the non-immunocompromised subjects (Figure 4a–c). No significant differences emerged between the median number of minor variants at T0 (N_I_ = 7, N_H_ = 9, *p* = 0.09, Mann–Whitney test), at T7 (N_I_ = 3, N_H_ = 3, *p* = 1, Mann–Whitney test) or overall (N_I_ = 16, N_H_ = 14, *p* = 0.08, Mann–Whitney test). For three immunocompromised and two non-immunocompromised subjects, it was possible to assess the longitudinal changes in the number of minor variants identified between T0 and T7(Figure 4d). Accordingly, although the number of the considered subjects was limited, the number of minor variants remained stable over the monitored timespan and variation seemed to be subject-dependent, regardless of the status of the immune system.

#### 3.4.3. Qualitative Analyses of Minor Variants

After demonstrating that the number of minor variants did not differ between immunocompromised and non-immunocompromised subjects, we assessed whether the type of mutations and their frequency were different in the two groups by performing a principal component analysis. Firstly, we ensured that, among the immunocompromised subjects, there was no difference in the minor variant profiles according to the SARS-CoV-2 variant of infection (Appendix A). Then, we investigated the impact of age (Appendix A) and treatments (Appendix A) on the profiles of minor variants. Finally, we generated a principal component analysis for all samples at T0, at T7 and regardless of the timepoints, considering the type and frequency of minor variants identified in each sample (Figure 5a–c). As a result, data did not cluster differently according to the immunological status. Although in some panels, such as panel a, most of the data seem to cluster in two different groups, the variance explained by the first two principal components is very low (less than 30%), further confirming the absence of a significant difference in the minor variant profiles of immunocompromised subjects compared to non-immunocompromised ones. Moreover, we monitored the conservation of low-frequency mutations over time in two immunocompromised patients to understand the dynamics of viral quasispecies composition. As shown in Figure 5d–e, the number of mutations fluctuates but is relatively stable between consecutive timepoints (grey dashed lines); nonetheless, there are always new mutations appearing (red lines in the plots) and replacing a portion of those that were present in previous samples (blue lines), suggesting that the quasispecies population undergoes intense renewal, especially in persistent infections.

## 4. Discussion

In this study, SARS-CoV-2 nasopharyngeal samples were collected at multiple timepoints from a group of eight immunocompromised subjects and from a group of fourteen non-immunocompromised subjects to investigate the impact of a compromised immune system on SARS-CoV-2 intra-host evolution. The immunocompromised group included subjects with haematological malignancies, subjects with solid tumours or organ transplant recipients receiving an immunosuppressive therapy. All the non-immunocompromised subjects received remdesivir therapy, whereas the immunocompromised subjects received either remdesivir or monoclonal antibodies or a combination thereof.

Each sample was deep-sequenced through the Twist SARS-CoV-2 Research Panel enrichment protocol to ensure a complete profiling of the target sequences and to obtain an unbiased representation of intra-sample variants. As a matter of fact, amplicon-based sequencing often leads to biased amplification across the genome due to differences in primer efficiency and affinity, especially in case of mismatches in the annealing regions [17,18]. More importantly, amplicon sequencing has been recently proven to provide a highly biased representation of minor allele frequencies [18]. Conversely, given that enrichment methods are based on a larger number of probes than amplicon-based methods and can tolerate up to 20% mismatches [30], capture-based methods are more robust and provide a reliable representation of intra-sample low-frequency variants [17,18]. Moreover, among the most popular SARS-CoV-2 enrichment panels available, we opted for the SARS-CoV-2-specific panel of Twist Bioscience, which was proven to be the most sensitive and efficient [19]. To further improve the accuracy of the enrichment method applied in the current study, the number of PCR amplification cycles, which is known to be a source of artefacts, was minimised to 10 cycles during library preparation and to 9 cycles after the probe-based enrichment, being comparable with the number of amplification cycles normally performed for metatranscriptomic sequencing [18].

As a result, we obtained a total of 52 consensus sequences from 21 subjects (no sequencing results were obtained for 1 patient out of the 22 initially enrolled), with 4 out of 8 immunocompromised subjects and 1 out of 13 non-immunocompromised subjects being infected with the Delta variant. The remaining subjects were infected with the Omicron variant. 

All the subjects for whom the complete SARS-CoV-2 consensus sequence was available for at least T0 and T7 were assessed for intra-host viral evolution, resulting in seven immunocompromised and seven non-immunocompromised subjects. Although, as expected, immunocompromised subjects showed a prolonged infection compared with non-immunocompromised subjects (21 and 14 days of median infection length in immunocompromised and non-immunocompromised subjects, respectively, *p* < 0.001, Mann–Whitney test), no differences in the amount or type of major mutations that emerged during the infection was observed between the two groups. Three immunocompromised subjects out of seven (42.85%) and two non-immunocompromised subjects out of seven (28.57%) acquired novel mutations during the viral infection. Specifically, two out of seven immunocompromised and two out of seven non-immunocompromised subjects acquired a novel mutation within 7 days, suggesting the length of infection to be the major driver for the intra-host accumulation of mutations. In particular, the subject H_9, who was treated with remdesivir, acquired a synonymous mutation in ORF3b, while the subject H_6, treated with remdesivir, acquired a synonymous mutation in ORF1ab, specifically within the gene of the RNA-dependent RNA polymerase, and a 9-nucleotide deletion, 510_518del, resulting in the deletion of three amino acids, NSP1_G82_V84del, and an amino acid substitution, NSP1_M85V. Interestingly, this deletion was detected in SARS-CoV-2 sequences deposited in GISAD with a fluctuating frequency that peaked at 1.76% on September 2023. Moreover, the same mutation was detected at low frequency in the immunocompromised subject I_5, who only received bamlanivimab–etesivimab treatment, and who carried five different deletions within the genomic region 508–524 which were competing at T14, with 519_524del becoming prevalent at T21. A previous work [29] defined the 500–532 Nsp1 locus as a deletion hotspot, with deletion variants being detected in 37 countries worldwide, and correlated such deletions to lower viral load and lower serum INF-β. Other two subjects acquired novel SARS-CoV-2 mutations during the monitoring carried out in this study: the subject I_4, who was treated with remdesivir, acquired two non-synonymous mutations, ORF1a:Q1519H and S: D420A. The ORF1a:Q1519H substitution has not been reported in the literature; however, it occurs within the gene of the Nsp3 protein. Nsp3 is the largest protein encoded by SARS-CoV-2, comprising up to 16 different domains and regions and it is essential for viral replication and transcription [31]. It participates in polyprotein processing, it interacts with the nucleocapsid protein and it binds viral RNA [31,32]. It has a critical role in counteracting host innate immunity as well, due to its de-ADP-ribosylating, de-ubiquitinating and de-ISGylating activities [31]. Finally, the subject I_7, who was treated with sotrovimab, acquired a silent mutation within the Nsp2 protein and a non-synonymous mutation within the ORF1a, T3182N, which lies in the Nsp3 protein, previously mentioned. It has been previously speculated that treatments might promote selective viral evolution. Specifically, the antiviral remdesivir has been suggested to foster viral evolution within the RNA-dependent RNA polymerase [8,9,15], whereas monoclonal antibody treatments are expected to promote the accumulation of escaping mutations within spike protein [28]. Overall, none of the subjects treated with mAb developed mutations within the spike gene region, whereas only one of the subjects treated with remdesivir, H_6, developed a synonymous mutation within the RNA-dependent RNA polymerase. Thus, the administered treatments did not affect the longitudinal acquisition of selective mutations in the assessed cohort. Moreover, no significant differences emerged in the number of subjects experiencing intra-host viral evolution nor in the amount of synonymous and non-synonymous mutations acquired within the infection from immunocompromised and non-immunocompromised subjects(Fischer exact test, *p* > 0.05). The longitudinal fixation of similar deletions within the same hotspot region, regardless of the immune system status and/or treatment received, further supports the hypothesis that the compromised immune system alone does not affect SARS-CoV-2 type or the number of novel mutations acquired.

We further explored the intra-host viral evolution by assessing the mutations occurring at low frequency. After validating and ensuring the reliability and consistency of the minor variants identified in replicated samples, we analysed all variants with a frequency ranging from 1% to 50% of samples showing a median coverage greater than 600 reads per genomic region. As a result, 30 out of the 52 samples were investigated for minor variants. Accordingly, no significant differences were observed in the amount and in the profile of minor variants identified in immunocompromised and non-immunocompromised subjects, nor between subjects infected with Delta or Omicron variants. Thus, minor variants analysis further supports the results of the consensus analysis, suggesting that the immune system state alone does not affect SARS-CoV-2 intra-host variability. Similarly, no clear impact of specific antiviral treatments was observed.

Another variable that could impair immune system competence is age, since older individuals could be affected by diminished immune responses [33]. We assessed this aspect by dividing the patients of our cohort into two groups, according to their age (≥65 vs. <65 years of age) instead of their immune status, and comparing the amount and type of minor variants between them. This analysis showed no difference either in the amount of minority variants between the two groups (Appendix A, N < 65 = 7, Mean < 65 = 127.3, N ≥ 65 = 11, Mean ≥ 65 = 105.5, *p* = 0.10, unpaired *t* test) or in the type of mutations (Appendix A, principal component analysis). However, individuals aged 65 and older exhibited slightly longer infection durations (*p* = 0.052, Mann–Whitney test, median < 65 = 21, IQR: 14–25.5; median ≥ 65 = 14, IQR: 14–14), indicating that age may have a comparable impact on SARS-CoV-2 infection as other more severe co-morbidities, albeit to a lesser extent. Such results agree with the literature by confirming that elderly subjects are more prone to experiencing prolonged SARS-CoV-2 infections, but maintain an adequate immune response compared to immunocompromised subjects [34].

The main limitations of this study are the low number of subjects recruited and successfully sequenced and the relatively short infection lengths observed in both groups, probably due to the SARS-CoV-2 variant of infection, and the effects of vaccines and antiviral treatments [35]. 

On the other hand, the study setting and methodologies chosen in this work diverge from most of the previous studies investigating the effects of immune system deficiencies on SARS-CoV-2 evolution [1,3,4,5]. In fact, all studies concerning low-frequency mutations differ for the choice of sequencing protocols and data analysis pipelines, with most of the works relying on amplicon sequencing, which has been reported to be strongly biased for low-frequency representation. Moreover, most of these studies lack a validation step, essential to calibrating low-frequency variant calling parameters, opting instead for arbitrary frequency and coverage thresholds. In addition, the majority of studies concentrate on individual immunocompromised patients experiencing unusually prolonged infections [36,37], representing exceptional cases that are challenging to compare with non-immunocompromised counterparts. Given that non-immunocompromised subjects generally do not experience severe symptoms or hospitalisation, persistent infections in such subjects are very rarely investigated [12,13,15,16], hindering a fair comparison with immunocompromised subjects and leading to a literature biased toward the latter. However, some research groups previously investigated SARS-CoV-2 intra-host evolution in immunocompetent subjects experiencing prolonged infection, reporting the longitudinal emergence of novel mutations, including immune-escaping mutations [12,15]. Similarly, other groups reported SARS-CoV-2 accumulation of mutations at the consensus level or at low frequencies in immunocompetent subjects in shorter time windows [13,16]. 

Overall, our data directly compare longitudinal intra-host SARS-CoV-2 diversity in immunocompromised and non-immunocompromised subjects, demonstrating that there are no disparities in the speed of new mutation emergence or in the type of mutations acquired between the two groups. Nonetheless, considering that infection length is significantly longer in immunocompromised patients, viral persistence in these hosts may offer the virus increased chances for evolution, resulting in an indirect effect of immune system deficiencies in favouring the accumulation of mutations over time.

## Figures and Tables

**Figure 1 viruses-16-00447-f001:**
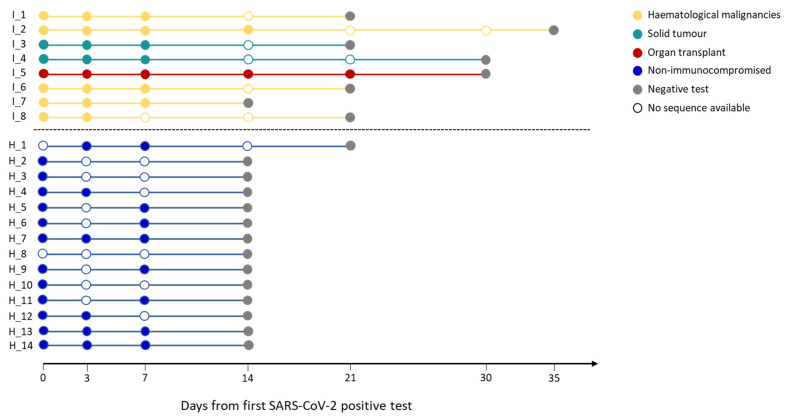
Infection length in immunocompromised and non-immunocompromised subjects. The figure reports the length of the SARS-CoV-2 infection in eight immunocompromised (Group 1) patients and in fourteen non-immunocompromised subjects at high risk of COVID-19 clinical progression (Group 2). The infection length is defined as the time window spanning from the first day of nasopharyngeal test positivity (day 0) to the first negative nasopharyngeal test. Group 1 subjects are reported in blue, whereas the different immunocompromised subjects are coloured according to the type of compromising condition, provided in the legend. The first negative test is presented as a grey dot. The other dots represent each day at which the subjects were tested, with filled dots indicating the availability of the viral sequence for that timepoint.

**Figure 2 viruses-16-00447-f002:**
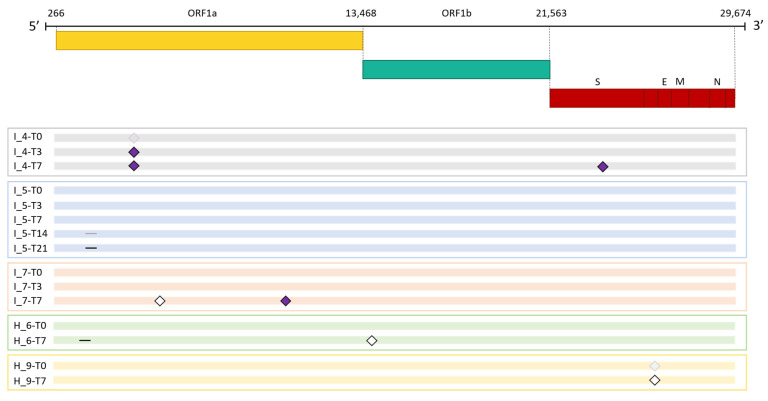
Emergence of SARS-CoV-2 consensus mutations in immunocompromised and non-immunocompromised subjects. The figure shows the longitudinal emergence of viral mutations in three immunocompromised (I) and in two non-immunocompromised (H) subjects. The longitudinal viral sequences of each patient are enclosed in a coloured box and are presented as coloured bars. Only the novel intra-host mutations compared to T0 are reported. Non-synonymous mutations are depicted as purple-filled diamonds, while silent mutations are presented as white diamonds. Deletions are indicated as black lines. Shaded mutations represent mutations present with a frequency below 50%.

**Figure 3 viruses-16-00447-f003:**
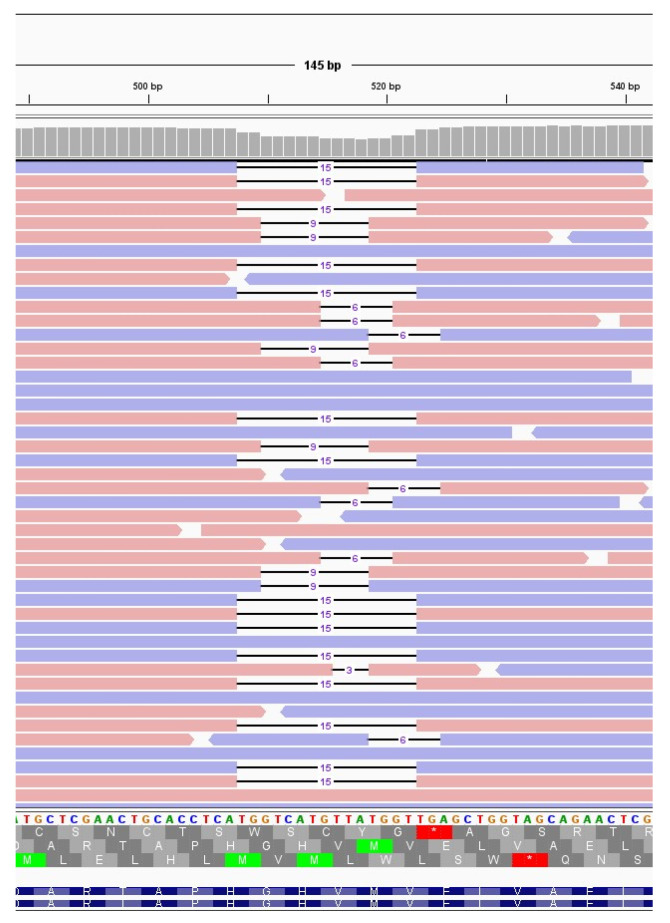
Deletions emerging in subject I_5 at T14. Visualisation of sequence reads spanning the SARS-CoV-2 genomic region 508–524 at T14 of the immunocompromised subject I_5. Reads mapping forward are reported in red, while reverse reads are depicted in blue. The black lines represent deletions, whose length is indicated in figures.

**Figure 4 viruses-16-00447-f004:**
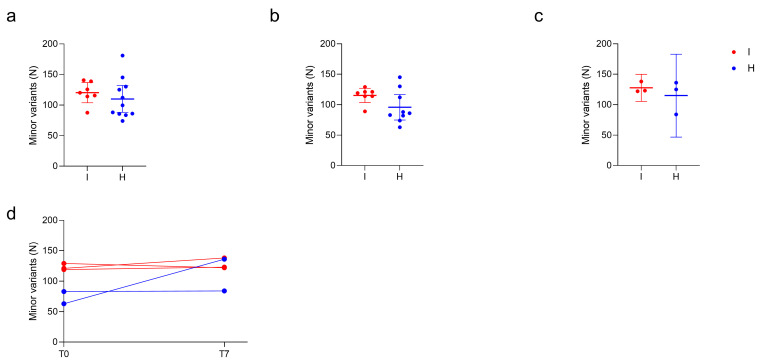
Quantitative analysis of minor variants. Comparison of the number of minor variants identified in immunocompromised (I) and in non-immunocompromised (H) subjects. (**a**) Comparison of the number of minor variants identified in immunocompromised and non-immunocompromised subjects, regardless of the timepoint at which the samples were collected (N_I_ = 7, Mean _I_ = 120.3, N_H_ = 11, Mean _H_ = 110. 0, *p* = 0.46, unpaired *t* test). (**b**) Comparison of the number of minor variants identified in immunocompromised and non-immunocompromised subjects at T0 (N_I_ = 7, Mean _I_ = 115.3, N_H_ = 9, Mean _H_ = 95.89, *p* = 0.10, unpaired *t* test). (**c**) Comparison of the number of minor variants identified in immunocompromised subjects and non-immunocompromised subjects at T7 (N_I_ = 3, Mean _I_ = 127.7, N_H_ = 3, Mean _H_ = 115.0, *p* = 0.49, unpaired *t* test). (**d**) Longitudinal intra-host variation of the number of minor variants identified in subjects for which T0 and T7 samples were available (N_I_ = 3, N_H_ = 2). The number of minor variants detected in each sample of an immunocompromised subjects are reported as red dots. Conversely, the number of minor variants detected in each sample of a non-immunocompromised subject are reported as blue dots, according to the legend.

**Figure 5 viruses-16-00447-f005:**
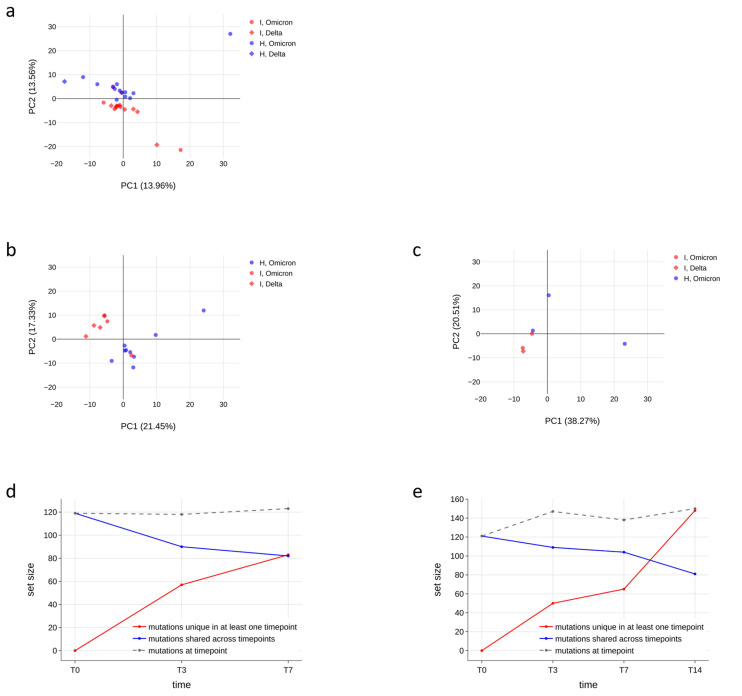
Minor variant profiles of immunocompromised and non-immunocompromised subjects. (**a**–**c**) Principal component analysis (PCA) of all the minor variants observed in samples collected from either the immunocompromised or the non-immunocompromised subjects at any timepoint (**a**); at T0 (**b**); and at T7 (**c**). According to the legend, minor variant profiles of immunocompromised subjects are provided as red dots, whereas the ones of non-immunocompromised subjects are presented as blue dots. Profiles of subjects infected with the Delta variant are represented as diamonds, whereas dots indicate Omicron as the variant of infection. The variation explained by each PC is provided along the relative axis. (**d**–**e**) Longitudinal analysis of the minor variants that persisted in all the considered timepoints (blue line), of the minor variants identified at least at one timepoint (red line) and of the minor variants specific to each specific timepoint in subject I_3 (**d**) and in subject I_5 (**e**).

**Table 1 viruses-16-00447-t001:** Features of the immunocompromised and the non-immunocompromised cohorts. Information regarding gender, age, type of compromising condition, treatments, vaccination status, Ct values and CD4/CD8 lymphocyte counts of immunocompromised subjects are reported in the table. Differences in gender and age between the two groups were tested with Fisher’s exact test. Differences in infection length were tested with Mann–Whitney test.

ID	Gender	Age	Compromising Condition	Infection Length	Monoclonal Ab	Antivirals	Vaccinated	Ct Values	CD4/CD8 Lymphocyte Counts
I_1	M	45	1	21	casirivimab–imdevimab	no	no	25/25/25	63/25
I_2	M	64	1	35	casirivimab–imdevimab	no	yes	18/20.5/17.6/23.2	64/32
I_3	F	69	2	21	casirivimab–imdevimab	remdesivir	no	14.5/16.7/26	260/57
I_4	M	71	2	30	no	remdesivir	yes	16.5/13/23	827/33
I_5	M	49	3	30	bamlanivimab–etesivimab	no	yes	17.6/16/15/26.9/31.8	243/27
I_6	F	69	1	21	sotrovimab	no	yes	12.4/23.8/16.8	152/24
I_7	M	55	1	21	sotrovimab	no	yes	23.6/29.9/30.8	192/62
I_8	M	58	2	14	no	remdesivir	yes	13.9/19.2	240/54
Total	8 (M = 75%)								
Mean		60							
Median				21					
H_1	M	37	8	21	no	remdesivir	yes	16.3/30.1	
H_2	M	76	5	14	no	remdesivir	yes	23.6	
H_3	M	70	6	14	no	remdesivir	yes	20.2	
H_4	F	82	5	14	no	remdesivir	yes	19.1/22.7	
H_5	M	74	4	14	no	remdesivir	yes	23.6/25.4	
H_6	M	83	5	14	no	remdesivir	yes	27.5/25.6	
H_7	F	59	6	14	no	remdesivir	yes	20.6/23.8/27.8	
H_8	F	69	4	14	no	remdesivir	yes		
H_9	M	89	5	14	no	remdesivir	yes	16.8/23	
H_10	F	66	7	14	no	remdesivir	yes	12.6	
H_11	F	73	4	14	no	remdesivir	yes	26/26	
H_12	F	85	5	14	no	remdesivir	yes	20.5/27.8	
H_13	F	48	6	14	no	remdesivir	yes	15.2/22.5/23.6	
H_14	M	93	5	14	no	remdesivir	yes	19.8/20.2/25	
Total	14 (M = 50%)								
Mean		71.7							
Median				14					
*p* value	ns	ns		<0.001					

1 = Haematological malignancies. 2 = Solid tumour. 3 = Organ transplant. 4 = Chronic lung disease. 5 = Heart disease. 6 = Chronic liver disease. 7 = Diabetes mellitus, type 2. 8 = Obesity (BMI > 30 kg/m^2^).

**Table 2 viruses-16-00447-t002:** SARS-CoV-2 evolution in immunocompromised and non-immunocompromised subjects. The longitudinal emergence of mutations at the consensus level is summarised for the seven immunocompromised and the seven non-immunocompromised subjects with complete genome sequences available for at least timepoints 0 and 7. For each subject, the viral strain identified, the number of acquired mutations, the nucleotide changes and the relative amino acid changes, if present, are reported.

ID	SARS-CoV-2 Lineage (PANGO)	Clade (WHO)	Evolved	Novel Mutations (N)	Timepoint of Novel Mutations	Deletions	Nucleotide Substitutions	Amino Acid Mutations
I_1	AY.98.1	Delta	no	-				
I_2	AY.43	Delta	no	-				
I_3	AY.101	Delta	no	-				
I_4	AY.43	Delta	yes	1	T3, T7		4822A > C, 22821A > C	ORF1a:Q1519H, S: D420A
I_5	BA.1	Omicron	yes	1	T21	519_524del		ORF1a: E87K, ORF1a: V84_M85del
I_6	BA.1.1.1	Omicron	no	-				
I_7	BA.1.1	Omicron	yes	2	T7		4012C > A, 9810C > A	silent, ORF1a: T3182N
Total			3 (42.85%)					
H_5	BA.2	Omicron	no	-				
H_6	BA.2	Omicron	yes	2	T7	510_518del	14602T > C	ORF1a: M85V, ORF1a: G82_V84del, silent
H_7	BA.1	Omicron	no	-				
H_9	BA.2.9	Omicron	yes	1	T7		25603T > C	silent
H_11	BA.1.1	Omicron	No	-				
H_13	BA.1.1	Omicron	No	-				
H_14	BA.1.1	Omicron	no	-				
Total			2 (28.57%)					
*p* value			ns					

**Table 3 viruses-16-00447-t003:** Low-frequency deletions emerging in subject I_5 at T14. List of low-frequency deletions observed in subject I_5 at T14. The deletion that became prevalent at the subsequent timepoint is depicted in bold. The median coverage of the genomic region 508–524 was 1698.5 reads per genomic position.

Deletion	Frequency (%)	Haplotype
508_522del	15.44	NSP1_G82del, NSP1_H83del, NSP1V84del, NSP1_M85del, NSP1_V86del
510_518del	8.20	NSP1_G82del, NSP1_H83del, NSP1V84del, NSP1_M85V
515_520del	6.49	
516_518del	1.24	
518_520del	1.57	
519_524del	6.67	NSP1_V84del, NSP1_M85del, NSP1_E87K

## Data Availability

The data presented in this study are openly available in GISAID repository at doi:10.55876/gis8.231130qo. The GISAID data availability table is available at https://github.com/MedCompUnipd/SARS-CoV-2_immunocompromised, accessed on 10 May 2023.

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
