# Peer review of "Immune System Deficiencies Do Not Alter SARS-CoV-2 Evolutionary Rate but Favour the Emergence of Mutations by Extending Viral Persistence"

_viruses, 2024, doi:10.3390/v16030447_

Round 1

Reviewer 1 Report (Previous Reviewer 1)

Comments and Suggestions for Authors

The authors have fully addressed my concerns.

Author Response

No need to reply

Reviewer 2 Report (Previous Reviewer 2)

Comments and Suggestions for Authors

The authors have addressed my concerns and this paper is suggested for publication.

Author Response

No need to reply

Reviewer 3 Report (New Reviewer)

Comments and Suggestions for Authors

The paper is interesting and rather well written

Below are some minor comments. The most missing part is comparison of obtained results with results previously obtained by other research groups

Library prep: (1) please, specify what was the input RNA amount? (2) “Adapted libraries” sounds strange (3) to me, 10 cycles of initial amplification is more or less typical. What polymerase did you use? 

Data analysis: probably it would be better to put the commands in the supplementary material.

Graphical representation of mutations can be improved

Please, discuss the comparison of your results with results obtained in other works on immunodeficient patients (for example, DOI: 10.1038/s41467-022-34033-x and DOI: 10.1080/21645515.2022.2101334).

Author Response

The detailed responses to reviewer's comments are present in the pdf attached.

This manuscript is a resubmission of an earlier submission. The following is a list of the peer review reports and author responses from that submission.

Round 1

Reviewer 1 Report

Comments and Suggestions for Authors

In the manuscript, Manuto and his colleagues explore the intra-host evolution of SARS-CoV-2 in immunocompromised patients. They gathered longitudinal nasopharyngeal swabs from 8 immunocompromised patients and 14 non-immunocompromised individuals infected with SARS-CoV-2, subsequently comparing viral evolution between these groups through deep sequencing. Their findings suggest that a compromised immune system does not significantly influence the within-host genomic variability of SARS-CoV-2. While the manuscript is articulate and clear, certain aspects undermine its overall conclusion and scientific impact:

1.     The small sample sizes of both the immunocompromised and non-immunocompromised groups limit the robustness of the statistical analysis, thereby weakening the primary conclusion of this study.

2.     In the non-immunocompromised group, the two subjects exhibiting intra-host viral evolution were aged 83 and 89. Given that older individuals can have diminished immune responses, the study should consider and discuss the potential age-related aspects of intra-host viral evolution of SARS-CoV-2.

3.     The varied antiviral treatments administered to individuals in the two cohorts could have influenced the study outcomes.

4.     Enhancing the manuscript with detailed information on the infected strains and viral titers in Table 1 would be beneficial and informative.

5.     In Figure 4, the comparison of the number of minor variants in immunocompromised and non-immunocompromised patients indicates no significant statistical differences. However, Figure 4a shows about a 1.4-fold increase in minor variants in immunocompromised ("I") subjects compared to non-immunocompromised ("H") subjects. The authors are encouraged to apply various statistical methods to validate the significance of this discrepancy.

Reviewer 2 Report

Comments and Suggestions for Authors

Review of Manuto et al.

In this study, Mediavilla et al collected nasopharyngeal swabs from 22 patients, positive to SARS-CoV-2 with immunocompromised and non-immunocompromised adult subjects. They deep sequenced the samples analyzed the data for longitudinal viral haplotype variation and for low frequency mutations diversity and shifts. This work is interesting and valuable, while the reviewer still has some concerns (see below) about this work, these issues need to be addressed before considering for publication.

1. The reviewer noticed that the subjects were under antiviral therapy. Do the authors know whether this may affect the immune system or viral replication in these patients? This at least should be discussed in the manuscript based on other publications.

2. Figure 5, please use the same font and size in this figure.

Comments on the Quality of English Language

N/A

Reviewer 3 Report

Comments and Suggestions for Authors

Previous research has highlighted the potential for SARS-CoV-2 mutations to occur through intrahost evolution, possibly linked to a host's immunocompromised state. This study posited that an immunosuppressed condition could increase the likelihood of prolonged SARS-CoV-2 infections, potentially facilitating intrahost evolution. However, the data showed comparable mutation patterns between 8 immunocompromised and 14 non-immunosuppressed subjects. While this study contributes to our understanding of SARS-CoV-2 mutation and viral evolution, several key aspects require additional clarification and revision.

Major Concerns:

1.   It remains ambiguous whether the immunocompromised subjects include HIV-1 positive patients, or are they limited to the diseases outlined in Table 1.

2.     The study should consider collecting and comparing samples before and after treatment to ascertain the treatment's impact on mutation selection.

3.     The differences in receptor binding capabilities between the Delta and Omicron variants might significantly influence virus replication and evolution, which warrants consideration in the study.

4.     The immunosuppressed subjects, afflicted with various diseases, might be undergoing diverse non-viral treatments, affecting their immunosuppressed status. It is crucial to know if the authors monitored their immune status and function.

5.     The methodology of collecting samples for longitudinal surveillance over       30 days may not be sufficient to detect viral mutation occurrence.

6. The study's limitations, including a low number of cases, short follow-up    duration, and variations in treatment and vaccination, might lead to the observed similarities in mutation patterns between immunocompromised and immunocompetent subjects. Consequently, it cannot be conclusively stated that a compromised immune system does not impact SARS-CoV-2 intrahost genomic variability